# Detection of Putative Mutation I873S in the Sodium Channel of *Megalurothrips usitatus* (Bagnall) Which May Be Associated with Pyrethroid Resistance

**DOI:** 10.3390/insects14040388

**Published:** 2023-04-17

**Authors:** Ruibo Gao, Rongcai Lu, Xinyao Qiu, Likui Wang, Kun Zhang, Shaoying Wu

**Affiliations:** 1Sanya Nanfan Research Institute, Hainan University, Yazhou, Sanya 572024, China; g493432710@163.com (R.G.); lrc18889600713@163.com (R.L.); m18846913713@163.com (X.Q.); 19937157228@163.com (L.W.); 2College of Plant Protection, Hainan University, Haikou 570228, China; 3Yazhou Bay Science and Technology City, Yazhou, Sanya 572024, China

**Keywords:** *Megalurothrips usitatus*, pyrethroids, resistance mutation, voltage-gated sodium channel

## Abstract

**Simple Summary:**

*Megalurothrips usitatus* is the main pest of cowpea in China, mainly by feeding on plant leaves and spreading plant viruses. In this study, a new mutation putative site, I873S, was correlated with pyrethroid resistance. According to the results of the bioassay, an I873S mutation may be associated with the resistance of *M. usitatus* to pyrethroids. The results of this study will promote the development of research on the resistance of *M. usitatus* to pyrethroids and provide guidance for improving the management system of *M. usitatus* in the future.

**Abstract:**

Pyrethroid resistance of thrips has been reported in many countries, and knockdown resistance (*kdr*) has been identified as a main mechanism against pyrethroids in many insects. To characterize pyrethroid resistance in *Megalurothrips usitatus* from the Hainan Province of China, we conducted a biological assay and sequenced the voltage-gated sodium channel gene domain II from *M. usitatus* field populations. It showed high resistance to the pyrethroids for 2019 and 2020, in which LC_50_ to lambda-cyhalothrin of *M. usitatus* was 1683.521 mg/L from Sanya in 2020. The LC_50_ value of deltamethrin was lower in Haikou than in other locations, which mean the south of Hainan has higher resistance than the north of Hainan. Two mutations of I873S and V1015M were detected in the domain II region of the sodium channel in *M. usitatus*; however, the mutation frequency of V1015M was only 3.33% and that of I873S was 100%. One is homozygous and the other is a heterozygous mutant type. The three thrips-sensitive strains of sodium channel 873 are highly conserved in amino acids (isoleucine), while the *M. usitatus* pyrethroid-resistant strains are all serine, so I873S may be related to the resistance of *M. usitatus* to pyrethroids. The present study will contribute to the understanding of the evolution of pyrethroids resistance and contribute to the development of resistance management of *M. usitatus* in Hainan.

## 1. Introduction

Cowpea, *Vigna unguiculata* L. Walp, is an economical and popular crop in the tropical and subtropical provinces of China [1,2]. Hainan Province is the “vegetable basket” of China, especially in winter. The quantity of cowpea increased from 2009 to 2019 in Hainan [3]. Thrips are the most serious pests of *V. unguiculata* [4,5]. A total of 296 species of thrips have been found in the world, and 97 species are known in China [6]. For the high average temperature in Hainan, the developmental duration of thrips is shortened and the generations overlap, causing them to break out in a short time [7]. The bean flower thrip *Megalurothrips usitatus* (Bagnall) (Thysanoptera: Thripidae) is widely distributed in cowpea, which causes serious damage in Hainan [4]. *M. usitatus* occurs every growing season and causes necrosis and premature dropping of buds and flowers due to its feeding and ovipositing; therefore, the value of the crop is reduced, resulting in yield losses [8].

It was shown that pyrethroids, neonicotinoids, spinosyns, and avermectins were used to control *M. usitatus* in the past 10 years (Table A1). From 2014 to 2017, the most commonly used insecticides in Hainan Province to control *M. usitatus* were beta-cypermethrin, emamectin benzoate, acetamiprid, spinetoram, abamectin, and imidacloprid (Appendix A Table A1; Appendix B Figure A1). The resistance of *M. usitatus* increased with the application of insecticides year by year. Moreover, *M. usitatus* had higher resistance to pyrethroids (LC_50_ = 32.74–131.02 mg/L) than the other five insecticides (1.98–75.52 mg/L) (Appendix B). Pyrethroids have been the most widely used neurotoxic insecticide in *V. unguiculata* for several decades.

Voltage-gated sodium channels play a very important role in the initiation and propagation of electrical signals in the nervous system [9]. The insect sodium channel is composed of an α subunit and several β subunits [10]. The α subunit consists of four homologous domains, and each domain is composed of six transmembrane domains [11]. The mutations of the sodium channel are the main reason for the resistance of pests to pyrethroids, which leads to a decrease in the affinity of pyrethroids to sodium channels [11,12]. The mutations causing resistance to pyrethroids occur in domain regions, especially in domains II and III [9]. According to previous studies, two pyrethroids binding domains were predicted. Pyrethroids binding site 1: domain IIS4–S5 binding peptide, IIS5 and IIIS6; pyrethroids binding site 2: domain IS4–S5 binding peptide, IIS5 and IIS6. Many mutations have been found in these regions [13,14]. The most classic mutations are 1014 and 918, which are currently found in insects with high levels of resistance [15,16,17]. *Kdr* resistance was first reported in *Musca domestica* [16,18,19], then it was reported in many pests [20,21,22,23,24]. The R957G mutation was found in the pollen beetle *Meligethes aeneus* [25], and the L899V and D953G mutations were found in *Cimex hemipterus* [26]. Other mutations also exist in domain II, among which M827I, I936V, L932F, L925I, V1010L, F1020S, and L1024V have been reported and verified in *Xenopus* oocytes [27,28,29,30,31,32,33]. Therefore, more than 60 *kdr* mutations or different combinations have been found in insects, and the sensitivity to pyrethroid agents is different, which better explains the interaction between pyrethroid and sodium ion channels at the molecular level [9].

There is still less research about the molecular mechanism underlying the resistance of *M. usitatus* to pyrethroids. In this study, domain II of the sodium channel of *M. usitatus* was cloned, and the mutations that might endow resistance to pyrethroid were determined by sequencing. The gene frequency of the obtained mutation was calculated in three field populations of *M. usitatus* from Hainan, China. Knowing these mutations may help us to quickly understand the resistance of *M. usitatus* to pyrethroids in the field. The discovery of these mutations provides a solid basis for our subsequent validation of the mechanism of *M. usitatus* pyrethroids’ resistance using electrophysiological assays.

## 2. Materials and Methods

### 2.1. Sample Collection

The fields of *M. usitatus* were collected from Haikou, Sanya, and Ledong in Hainan Province (Figure 1). They were propagated in the laboratory to prepare for the follow-up experiment. The relatively susceptible strain of *M. usitatus* (HNS) were fed with cowpea (No. 2 Xia Bao, Xingning city, Guangdong, China) grown and planted without using of any insecticides in a light incubator (RXZ-380BM, Ningbo Yanghui Instrument Co., Ltd., Ningbo, China) with conditions (26 ± 2) °C, 50–60% relative humidity, 16:8 h (L:D). The cowpeas were cut into 8–10 cm lengths with solid ends. The cowpeas were put in a transparent tissue culture bottle with spawning tablets on the bottom and the bottle was filled without any insecticide beans (8–10 cm). A large square hole was cut on the top of the tissue culture bottle. A 200-mesh screen was installed and fastened with a leather band to prevent the thrips from drilling out and escaping.

### 2.2. Insecticide Bioassay 

Lambda-cyhalothrin and deltamethrin were bought from Nutrichem Company Limited Co., Ltd., Beijing, China. The purities of lambda-cyhalothrin and deltamethrin were 95% and 97.5%, respectively. The toxicity levels of lambda-cyhalothrin and deltamethrin in adults of thrips were determined with a thrips insecticide bioassay system (TIBS) [34]. Briefly, the insecticide was diluted with 0.1% Triton-water (Sigma, St. Louis, MO, USA) into eight concentrations, and each concentration was repeated three times. The insecticides were added into the 1.5 mL centrifuge tube and the insecticides were poured upside down several times, and the centrifuge tube was placed in a ventilated place to dry. The non-toxic beans were cut into small segments of approximately 1 cm and soaked in the insecticides for 15 s. Then, the surface of the beans was air-dried and they were placed in a centrifuge tube. The lambda-cyhalothrin and deltamethrin were dissolved with a purity of 95% and 97.5% into 20 mL of acetone solution, and the highest concentration of 1600 mg/L was configured to weigh 33.7 and 32.8 mg of lambda-cyhalothrin and deltamethrin. The prepared mother liquor was diluted to the concentrations of 800, 400, 200, 100, 50, 25, and 12.5 mg/L with 0.1% Triton water, and treated in contrast with 0.1% Triton water. For susceptible populations of *M. usitatus*, we needed to weigh 0.2 mg of lambda-cyhalothrin and deltamethrin to configure the highest concentration of 10 mg/L, and still dilute to 5, 2.5, 1.25, 0.625, 0.3125, and 0 mg/L with 0.1% Triton water. A self-made trematode tube was used to collect 20 adult thrips, which were then placed into the centrifuge tube. The top of the tube was covered with a 200-mesh screen and fixed with a leather band. LC_50_ values were calculated using PoloPlus 2.0 software (LeOra Software Inc., Petaluma, CA, USA). The data were analyzed via the Probit model and a Chi-square test was performed to obtain the *p* value, using SPSS software version 26.0 (IBM SPSS Statistics, International Business Machines Corporation, Armonk, NY, USA). LC_50_ values were considered to be significantly different if their 95% FLs did not overlap.

### 2.3. DNA Extraction and Sodium Channel Cloning of M. usitatus 

The DNA of individual thrips was extracted by using the kit of a medium/large number of cells/tissue genomic DNA Extraction Kit (Biotake Co., Ltd., Beijing, China). The individual *M. usitatus* used to extract DNA are all field populations without bioassay. The improved DNA extraction steps are as follows: First, the individual thrips were put into a 1.5 mL centrifuge tube and freeze them with liquid nitrogen for 10 min. In detail, after sample homogenization, 50 µL of cell nuclear lysate was added to the solution and mixed. The solution was incubated at 65 °C for 30 min. Then, 17 µL of protein precipitation solution was added, shaken for 30 s, incubated for 5 min on ice, and centrifuged at 13,000 rpm at room temperature for 13 min. A total of 50 µL of the supernatant was carefully transferred to a new 1.5 mL centrifuge tube and the same amount of isopropanol was added, mixed well, and incubated at 4 °C for 1 h. After incubation, centrifuge at 10,000 rpm at room temperature for 6 min, carefully remove the supernatant, add 50 µL 67% alcohol, mix well, and centrifuge at 10,000 rpm at room temperature for 6 min. Carefully remove the 67% alcohol and dry the 10 min in the ultra-clean table, dissolving the DNA in the 10 µL Elution buffer. The experimental reagents used in our DNA extraction scheme are from Biotake Biotechnology Co., Ltd, Beijing, China. The DNA concentration was measured using a NanoDrop2000 spectrophotometer (ThermoFisher, Waltham, MA, USA), verified by 1% agarose gel electrophoresis, and stored at −20 °C. The paired primers of *M. usitatus* for the sodium channels of domain II were designed based on the sequences of the sodium channels of *Frankliniella occidentalis*, *Thrips palmi*, and *T. tabaci* (GenBank accession numbers: XM_026421466.1, AB849921.1, and LC389826.1, respectively). The primer sequences were forward: 5′-AAGAAGACGAGGAGGATCCTACG-3′ and reverse: 5′-GTTAGTGTAACTGTCTGCCGTG-3′. PCR amplification was carried out as follows: one cycle of 94 °C for 1 min; 36 cycles of 94 °C for 15 s, 56 °C for 30 s, and 72 °C for 1 min; and one cycle of 72 °C for 10 min. The PCR products were tested by 1% agarose gel electrophoresis. The PCR products of *M. usitatus* were purified and collected. The purified product was linked to the T vector (Nanjing Vazyme Biotech Co., Ltd., Nanjing, China) and introduced into stbl2 (Shanghai Weidi Biotechnology Co., Ltd., Shanghai, China) for transformation. The plasmid of the positive clone was extracted and sequenced by Shanghai Boshang Biotechnology Co., Ltd. (Shanghai, China). Then, SNP/INDEL was used to analyze the pyrethroid resistance-related mutations in the *M. usitatus* sodium channel gene of domain II. The qualified DNA-Seq reads of the 30 biological replicates of each *M. usitatus* population were mapped to the cloned partial Nav channel gene sequence of the Haikou, Ledong, and Sanya field population using Bowtie2.2.2.6. Single nucleotide polymorphism/insertion-deletion (SNP/INDEL) calling was performed with SAM tools 0.1.19 using the mapping results and then mutations and their genotype were identified in each replicate by using in-house Perl scripts [35].

### 2.4. Phylogenetic Analysis

The full length of the Na_v_ sequences of 34 insects were downloaded from NCBI, and domain II was aligned. Subsequently, MEGA 7.0 was used to construct the molecular phylogenetic tree. First ClustalW was performed to compare the amino acid sequences of all insects, followed by neighbor joining (NJ) to construct the evolutionary tree, p-distance as the amino acid substitution model, pairwise deletion as the null/deletion data processing method, and bootstrap 1000 times. Finally, the images were processed using Photoshop. The NCBI accession numbers used to construct the species used in the *M. usitatus* evolutionary tree were as follows: *M. usitatus* (UOV21262.1), *T. palmi* (XP_034241363.1), *F. occidentalis* (XP_026278649.1), *Bombus impatiens* (ARH02610.1), *Bombus terrestris* (XP_020720206.2), *Apis mellifera* (AMB38675.1), *Apis dorsata* (XP_006613060.1), *Apis florea* (XP_031775133.1), *Apis cerana* (XP_016908431.1), *Dufourea novaeangliae* (XP_015436344.1), *Eufriesea mexicana* (XP_017755818.1), *Habropoda laboriosa* (XP_017797364.1), *Megachile rotundata* (XP_012144116.1), *Athalia rosae* (XP_025602780.1), *Polistes dominula* (XP_015185107.1), *Polistes canadensis* (XP_014608517.1), *Harpegnathos saltator* (XP_025152406.1), *Linepithema humile* (XP_012235106.1), *Monomorium pharaonis* (XP_012540566.1), *Wasmannia auropunctata* (XP_011689749.1), *Dinoponera quadriceps* (XP_014472746.1), *Acromyrmex echinatior* (XP_011061306.1), *Camponotus floridanus* (XP_025269367.1), *Pogonomyrmex barbatus* (XP_011636383.1), *Ooceraea biroi* (XP_026827719.1), *Solenopsis invicta* (XP_039304560.1), *Anopheles gambiae* (CAM12801.1), *Drosophila melanogaster* (AAB59195.1), *M. domestica* (NP_001273814.1), *Bombyx mori* (NP_001136084.1), *Heliothis virescens* (AAC26513.1), *Plutella xylostella* (XP_037971801.1), *Blattella germanica* (BBD13275.1), and *Tribolium castaneum* (QPF20448.1). Different colors were used to distinguish different orders of insects.

### 2.5. Identification of Mutation Sites in the Para Domain II

To check for possible site mutations associated with pyrethroid resistance, template DNAs from the *M. usitatus* field strains were amplified, and the identified nucleotide sequences were analyzed. The frequency of pyrethroid resistance-associated site mutations was examined using individual gene amplification and sequencing. The presence of pyrethroid resistance-associated site mutations in the field populations was determined using population gene amplification and sequencing.

## 3. Results

### 3.1. Toxicity of Lambda-Cyhalothrin and Deltamethrin on M. usitatus

Results on the toxicity of lambda-cyhalothrin and deltamethrin to *M. usitatus* are shown in Table 1 from Haikou, Ledong, and Sanya populations. Compared with the susceptible strain, the three field strains developed increased resistance to lambda-cyhalothrin 414- to 1128-fold. The deltamethrin treatment group showed a lower resistance ratio, which was 161- to 352-fold compared with the susceptible strain. The LC_50_ values of the Sanya population in 2020 for lambda-cyhalothrin and deltamethrin were the highest, reaching 1683.521 and 695.323 mg/L, respectively. The LC_50_ for 2020 in the three locations were all slightly higher than those for 2019, indicating that the resistance in these sites may be increasing annually. The LC_50_ values of three populations for lambda-cyhalothrin in 2019 and 2020 were slightly higher than those for deltamethrin, which indicated that the insect had higher resistance to lambda-cyhalothrin than deltamethrin.

### 3.2. Sequence Analysis of Sodium Channel Gene of M. usitatus

Based on the sequences of sodium channels of *F. occidentalis*, *T. palmi*, and *T. tabaci*, the sodium channel domain II of *M. usitatus* was cloned. From the perspective of the phylogenetic tree, the homology between *M. usitatus* and *T. palmi* in domain II was the highest, followed by that between *M. usitatus* and *F. occidentalis* (Figure 2). Domain II contains 681 nucleotides and translates 227 amino acids. 

### 3.3. Identification of Mutations of Domain II of M. usitatus to Pyrethroids

There were two base transitions of domain II from *M. usitatus*, which are I873 and V1015 (named according to the drosophila melanogaster sodium channel amino acid number). The first base is ATT-AGT transition in the domain II S3 region and the second base is GTG-ATG transition in the connecting region of domains II S6, compared with the susceptible sequences *M. usitatus* (Figure 3; Appendix C Figure A2). The V1015M mutation is located in the binding region of the pyrethroid (Figure 3A). Compared to the HNS population, the two mutations are I873S and V1015M (Appendix D Figure A3). In addition to these two mutation sites, we also detected the 918 and 1014 of the sodium channel of *M. usitatus* in the field. The results showed that no mutation was found at sites 918 and 1014 in the field population (Figure 3A; Figure A3).

### 3.4. Identification of Mutation Frequency 

The sodium channel domain II of *M. usitatus* were sequenced to determine the mutation frequency from Haikou, Ledong, and Sanya in 2019 and 2020 (Figure 3A). Four mutations in three populations of *M. usitatus* were monitored. The number of individuals of each population was 30 to 40, and about 200 individuals of *M. usitatus* were obtained totally. The genotypes of each sample were determined, and genotype frequencies in different populations were calculated (Table 2). Both in 2019 and 2020, all samples showed I873S mutations, only a 3.33% mutation frequency was V1015M, and 918 and 1014 mutation frequency were 0. Only one sample showed V1015M mutation in Sanya for 2020, not 2019, and no V1015M mutations were found in the Haikou and Ledong population in 2019 and 2020. The mutation frequency of I873S was 100% in Haikou, Ledong and Sanya populations. One is homozygous and the other is heterozygous mutant type (Table 2; Figure 3B).

## 4. Discussion

Hainan has superior hydrothermal conditions across the tropics and subtropics, and the occurrence of pests becomes serious with high temperature [1]. The main insecticides used to control *M. usitatus* have been presented in Hainan for the past decade (Appendix A). The number of thrips decreased after insecticides treatment, However, the resistance of *M. usitatus* to insecticides increased, including pyrethroids [36].

Sanya is located in the south of Hainan, which is more suitable for the growth and development of pests. Our resistance monitoring data showed that the Sanya population of *M. usitatus* has evolved higher lambda-cyhalothrin and deltamethrin resistance than the Haikou and Ledong populations. Tang reported the toxicities of β-cypermethrin, cyhalothrin, and bifenthrin in Haikou field population of *M. usitatus*, which exhibited a high LC_50_ value between 30.216 mg/L and 114.084 mg/L in 2014 [37]. The resistance of *M. usitatus* to β-cypermethrin in Sanya, Chengmai and Haikou from 2014 to 2015 was also checked, which showed the Sanya population evolved 8.07- and 12.99-fold resistance to β-cypermethrin [38]. Subsequent resistance monitoring showed that the resistance of *M. usitatus* to β-cypermethrin continued to increase from 2016 to 2017, and the Sanya population was 17.06- and 27.03-fold resistance, respectively [39]. In the present study, three field populations of *M. usitatus* showed high resistance to lambda-cyhalothrin and deltamethrin from 2019 to 2020 in Hainan, of which the highest LC_50_ value is 1683.521 mg/L. The resistance of thrips to pyrethroids increased year by year in Hainan. Meanwhile, resistance to other insecticides was also on the rise [37,38,39]. Sodium channels are the targets of pyrethroid resistance of insects. Insects exhibiting *kdr* have reduced sensitivity to pyrethroids and DDT resulting from one or more mutations in the insect sodium channel gene [10]. In 2014, more than 60 mutations related to pyrethroid resistance were found, and the number is still increasing in the world [11]. There are potentially many resistance-associated mutations in the four homologous domains of the sodium channel [40]. Furthermore, most of the mutations to pyrethroids were found to occur in the S4–S5, S5, and S6 segments of the domain II region of the channel protein [41]. 

Pyrethroid resistance has been reported in thrips, such as *T. tabaci*, *F. occidentalis*, and *T. palmi* [20,35,42]. The M918T, L1014F, and T929I had been reported in several pyrethroid-resistant *T. tabaci* strains in Japan [21]. The single mutations M918L and V1010A have been reported in pyrethroid-resistant *T. tabaci* populations in the United States [43]. Then, the other pyrethroid resistance-related mutations have been identified in *T. tabaci*, including the double mutation of T929I/K1774N and single mutation M918L in Japan [36]. Mutations M918T, T929I, and L1014F have resulted in pyrethroid resistance in *T. palmi* [44]. *F. occidentalis* demonstrated resistance due to L1014F and T929C mutations in the sodium channel [20]. Most mutations of thrips were founded in domain II. In this study, we detected domain II of sodium channels from the field populations. Two single mutations were found in domain II, namely, I873S and V1015M. Among the tested samples, the I873S was homozygous mutation. The mutation frequency of I873S was 100%, while that of V1015M was only 3.33%. As reported in the literature, insect sodium channel amino acid sequences are highly conserved [45]. In addition to the thrips-sensitive populations’ sodium channels 918 and 1014, which are both methionine and leucine, respectively, these two amino acids were also found in sensitive populations of other pests such as *Plutella xylostella* and *Myzus persicae* [11]. In addition, we also found that 873 in the sodium channel of all three thrips-sensitive populations is isoleucine, showing that the I873S mutation may be involved in pyrethroid resistance. There are no mutations on 918 and 1014 for *M. usitatus*, not like other thrips. Mutations associated with pyrethroid resistance could occur in 1014F almost in all of the insects; thus, it is beyond expectation. Lee et al. [46] proposed that the absence of a single M918T mutation in the resistant strain was due to the loss of function associated with mutation, and is somehow complemented or rescued by the presence of the L1014F mutation. It was hypothesized that in *kdr* populations that already carried the L1014F mutation, the *super-kdr* allele derived from M918T might be substituted as a selection for the second site mutation. Therefore, neither M918T nor L1014F was found in the sodium channel of *M. usitatus*, which may be related to the reproductive mode and mutation site characteristics of the population.

Could I873S of *M. usitatus* reduce sensitivity to pyrethroids for the mutation in all of filed individuals? V1015M and 1014F are close to each other, so, do the V1015M help I873S like other double mutations for resistance? In previous studies, the *super*-*kdr* mutation T929I binds to M827I on domain IIS1 and IIS2 linker and L932F on domain IIS5, which can completely disable the sensitivity of sodium channels to permethrin. In the linking peptide sequence of domain I and II found in *B. germanica*, two mutations, E435K and C785R, co-existed with L1014F, and the channel sensitivity was reduced by a factor of 100-fold when the E435K or C785R mutation combined with L1014F, while the simultaneous presence of the three mutations reduced the channel sensitivity to deltamethrin by a factor of 500-fold. They also found in the field population of *Aedes aegypti*, S989P and D1763Y mutations often coexisted with V1016G. Electrophysiological results showed that both V1016G/S989P and V1016G/D1763Y were more resistant to pyrethroids than single mutants. I873S was located outside the pyrethroid binding site of sodium channels. In addition, V1015M is close to L1014F and V1016G, so we suspect that the double mutation formed by them will significantly reduce the sensitivity of sodium channels to pyrethroids, resulting in pyrethroid resistance of *M. usitatus* [14,27,47]. Several studies have successfully confirmed the mechanism of other mutations (L1014F, T929I, and M918T) in *Xenopus* oocytes using electrophysiological techniques [48,49,50]. The function of the I873S and V1015M should be verified by the electrophysiological system in the future.

## 5. Conclusions

This is the first report of mutations I873S and V1015M in *M. usitatus* from Hainan province, I873S exists in homozygous mutant type in all three field populations and the bioassay results and SNP/INDEL analysis showed that I873S mutation may reduce the sensitivity of *M. usitatus* to pyrethroids. The information is valuable for monitoring the frequency of pyrethroid resistance in field pest populations as part of pest resistance management. At the same time, it is also a rapid method to identify sodium channel mutations in *M. usitatus.* Future research is needed to study the specific mechanism underlying the pyrethroid resistance of these two mutations. Molecular experiments were used to monitor the development of pyrethroid resistance of *M. usitatus* and to improve the resistance management system of *M. usitatus*.

## Figures and Tables

**Figure 1 insects-14-00388-f001:**
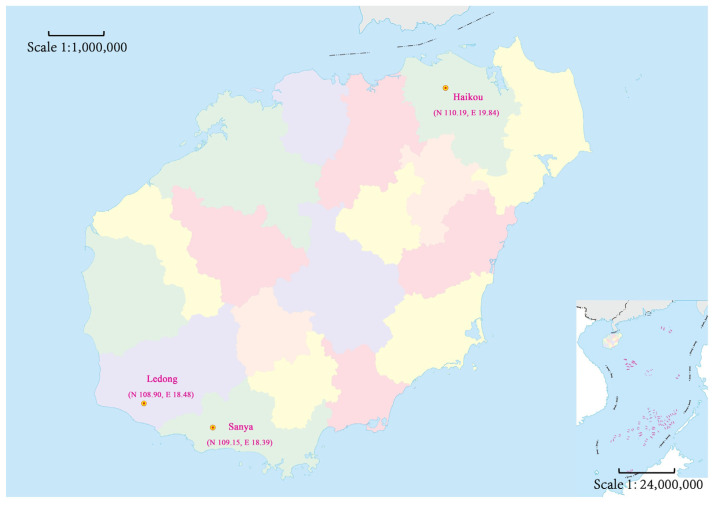
Map of Hainan Province showing the *M. usitatus* collection sites. (Supervised by Hainan Administration of Surveying Mapping and Geoinformation.) The scale of the main drawing is 1:1,000,000. In the lower-right corner is the South China Sea and the South China Sea Islands, with a scale of 1:24,000,000.

**Figure 2 insects-14-00388-f002:**
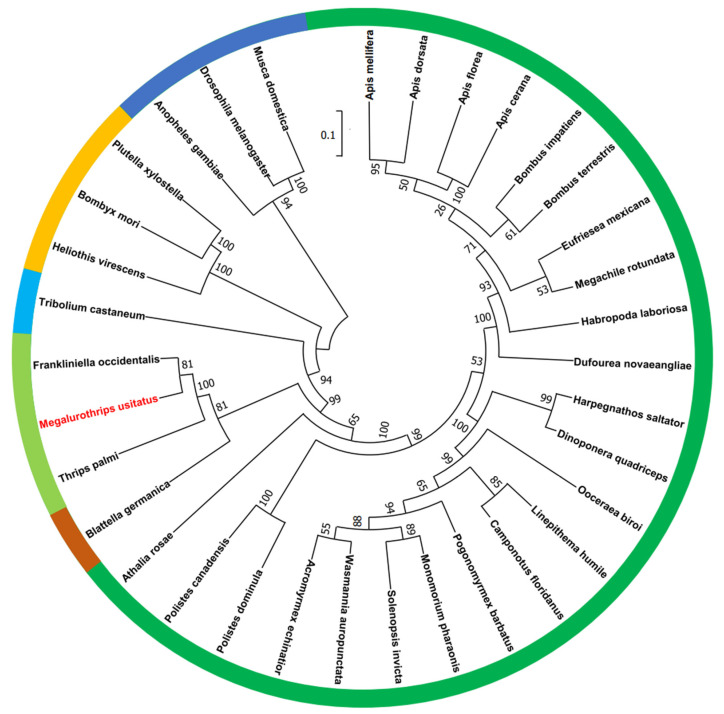
Phylogenetic tree of the insect sodium channel family. Dark green: hymenoptera; light green: thysanoptera; dark blue: diptera; light blue: coleoptera; orange: lepidoptera; brown: blattaria. The insects marked in red are the test insects *M. usitatus*.

**Figure 3 insects-14-00388-f003:**
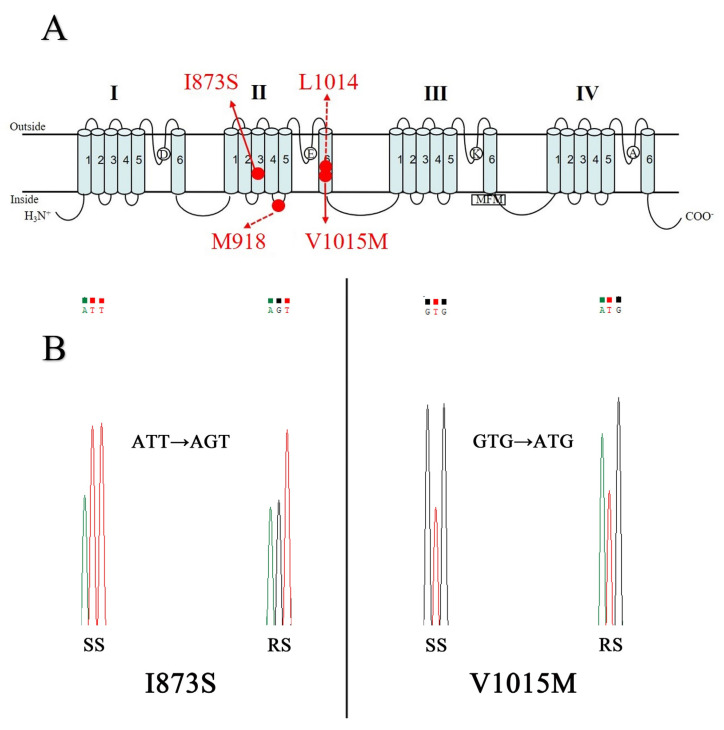
(**A**) Mutation site of domain II in *M. usitatus*. (**B**) Partial nucleotide sequence chromatograms of amino acids at positions 873 and 1015 of the sodium channel in *M. usitatus* susceptible and resistant strains. Amino acid substitution sites are boxed. Left: the nucleotide changes at site 873. Right: the nucleotide changes at site 1015.

**Table 1 insects-14-00388-t001:** Toxicity of lambda-cyhalothrin and deltamethrin to populations of *M. usitatus* in 2019–2020.

Insecticide	Year	Population	N ^a^	Slope ± SEM ^b^	LC_50_ (95% FL ^c^) (mg/L)	Chi-Squared	*p* ^d^	df	RR ^e^
Lambda-cyhalothrin		Susceptible population	360	1.204 ± 1.827	1.492 (1.204–1.827)	25.756	0.211	5	1
	2019	Haikou	380	1.368 ± 0.130	617.753 (479.626–807.666)	5.274	0.720	6	414
	2020		383	1.632 ± 0.229	1166.426 (896.381–1683.527)	15.467	0.289	6	782
	2019	Ledong	376	1.233 ± 0.130	902.456 (679.613–1251.270)	13.385	0.194	6	605
	2020		376	1.152 ± 0.130	1171.423 (857.377–1715.986)	11.764	0.732	6	785
	2019	Sanya	376	0.982 ± 0.118	1377.514 (960.220–2211.041)	9.29	0.336	6	923
	2020		420	0.804 ± 0.122	1683.521 (981.776–4082.404)	6.019	0.475	6	1128
Deltamethrin		Susceptible population	360	2.048 ± 0.183	1.976 (1.586–2.461)	20.756	0.279	5	1
	2019	Haikou	348	1.422 ± 0.160	317.335 (238.852–424.582)	18.171	0.257	6	161
	2020		373	2.124 ± 0.219	439.950 (354.242–552.580)	19.940	0.211	6	223
	2019	Ledong	348	1.596 ± 0.162	376.314 (303.747–473.193)	7.358	0.599	6	190
	2020		348	1.411 ± 0.158	514.285 (403.624–682.587)	6.175	0.553	6	260
	2019	Sanya	348	1.441 ± 0.160	526.441 (414.539–696.554)	6.091	0.772	6	266
	2020		348	1.287 ± 0.158	695.323 (526.472–994.208)	7.229	0.263	6	352

^a^ Number of thrips tested. ^b^ Standard error. ^c^ Fiducial limit. ^d^ *p* values are based on Chi-square goodness of fit test. *p* values > 0.05 suggest goodness of fit of the model. ^e^ RR (Resistance ratio) = LC_50_ of field population/LC_50_ of susceptible population.

**Table 2 insects-14-00388-t002:** Allele frequencies of I873S, M918, L1014 and V1015M mutant genotypes in six field populations.

Population	Year	Number of Insects	Nucleotide Change	Amino Acid Mutation	Mutation in Region	Mutation Type	Mutation Frequency
Haikou	2019	30	T-G	I873S	DIIS3	homozygous mutant type	100%
	2020	36	T-G	I873S	DIIS3	homozygous mutant type	100%
Ledong	2019	32	T-G	I873S	DIIS3	homozygous mutant type	100%
	2020	30	T-G	I873S	DIIS3	homozygous mutant type	100%
Sanya	2019	37	T-G	I873S	DIIS3	homozygous mutant type	100%
	2020	30	T-G	I873S	DIIS3	homozygous mutant type	100%
Haikou	2019	30	G-A	V1015M	DIIS6	heterozygous mutant type	0%
	2020	36	G-A	V1015M	DIIS6	heterozygous mutant type	0%
Ledong	2019	32	G-A	V1015M	DIIS6	heterozygous mutant type	0%
	2020	30	G-A	V1015M	DIIS6	heterozygous mutant type	0%
Sanya	2019	37	G-A	V1015M	DIIS6	heterozygous mutant type	0%
	2020	30	G-A	V1015M	DIIS6	heterozygous mutant type	3.33%
Haikou	2019	30	T-C	M918	DIIS4-S5linker	\	0%
	2020	36	T-C	M918	DIIS4-S5linker	\	0%
Ledong	2019	32	T-C	M918	DIIS4-S5linker	\	0%
	2020	30	T-C	M918	DIIS4-S5linker	\	0%
Sanya	2019	37	T-C	M918	DIIS4-S5linker	\	0%
	2020	30	T-C	M918	DIIS4-S5linker	\	0%
Haikou	2019	30	G-T	L1014	DIIS6	\	0%
	2020	36	G-T	L1014	DIIS6	\	0%
Ledong	2019	32	G-T	L1014	DIIS6	\	0%
	2020	30	G-T	L1014	DIIS6	\	0%
Sanya	2019	37	G-T	L1014	DIIS6	\	0%
	2020	30	G-T	L1014	DIIS6	\	0%

## Data Availability

The data presented in this study are available from the corresponding author upon reasonable request.

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
