# Peer review of "Detection of Putative Mutation I873S in the Sodium Channel of Megalurothrips usitatus (Bagnall) Which May Be Associated with Pyrethroid Resistance"

_insects, 2023, doi:10.3390/insects14040388_

Round 1

Reviewer 1 Report

Megalurothrips usitatus is a major pest of cowpea in China, which has evolved high resistance to insecticides, especially to pyrethroids. This study presents the resistance detection data to two pyrethroids of three populations collected in 2019~2020, and the detection data of the mutation frequency of the resistance gene Nav. This study provides information for further resistance research and management of this pest. The experiments were well designed and conducted. This manuscript was well written. I think this manuscript could be accepted after minor revisions.

1. In the manuscript, we cannot find a strong evidence to support that I873S mutation is related to the resistance of M. usitatus to pyrethroids. Although the mutation frequency of I873S was 100% in three field populations that have high resistance to lambda-cyhalothrin and deltamethrin, the susceptible strain and the detected field populations have different origins and different genetic backgroud. Without further experiment evidences like electrophysiological system in the current manuscript, this conclusion is not rigorous. So I think the title and related conclusions must be rewritten.

2. Readers might be also interested in the resistance data to other insecticides in M. usitatus. So it is better to move "Appendix B" to the "Results" section and add the related method description in the "Materials and Methods" section.

3. Please add more information about the figures and tables. Such as in "Table 1", add the values of degrees of freedom after chi-squared values and add the judgement whether the toxicity data fits to the probit model. In "figure 2", it is better to add more information about the constuction method and present description such as the means of different color signs.

3. Were the tested insecticide solutions prepared from technical samples with 0.1% Triton-water directly? These insecticides are hard to be dissolved in water. No organic solvent used? If using organic solvent, the control solution should contain the equivalent organic solvent to the highest insecticide solution.

4. Please add more description about the homozygous mutant detection in the "Materials and Methods" section.

5. There are some typing errors, such as the italic Latin insect names.

Reviewer 2 Report

This author monitored the resistance of six field populations of Megalurothrips usitatus to lambda-cyhalothrin and deltamethrin in Hainan Province, detected the mutation sites in the sodium channel domain II of M. usitatus, and found two new sites (I873S and L1015M). Among them, mutation I873S may be associated with population pyrethroids insecticide resistance. This study can establish a molecular method for rapid detection of pyrethroid resistance in field population of M. usitatus, which has great significance for the resistance management of M. usitatus. However, this article needs to be revised before it is published in insects.

1. The “mutations” in the title should be “mutation”, and Megalurothrips usitatus Latin lacks nomenclators., please revise the title.

2. L28, L263 and table2, The V1015M mutation frequency is only 3.33%, and I have doubts that it is a homozygous mutation. L29-31, both pyrethroids resistance evolution and resistance management practices were targeted at thrips rather than Hainan winter vegetables.

3. L39-40, Data lacked reference support.

4. L47, spinosads are biological insecticide, which are duplicated with the later biological insecticides, and are also wrong in Appendix A. Appendix A of Imidacloprid not belong to Diamide, and Emamectin-benzoate not belong to biological insecticide.

5. L56, “Sodium channels” should modify to “Voltage-gated sodium channels”. L75, “understand” should modify to “quickly understand”.

6. The abbreviated symbols of latitude and longitude of the collection sites (N, E) should be added to Figure 1.

7. L97, Whether the authors can prove that Triton has no effect on thrips. It is suggested to change the America to the USA in the whole text, for example L97and L111.

8. L113, “a single” should modify to “individual”. L116, “single thrips” should modify to “individual thrip”. L119, “Protein” should modify to “protein”. L128, “ThermoFisher” is not Chinese.

9. L192, The figure notes for Figure 2 should be labeled with what the colors represent.

10. There is a formatting error in the full text that requires Spaces, for example, L119 “30min”, L120 “5min”, L121 “13min” “50µL” “1.5mL” ...... and so on.

11. L143, Doing the phylogenetic tree uses 34 insects, not 36.

12. In the discussion, L265-267, The authors should add some more discussion of why no mutations were detected in the Megalurothrips usitatus sodium channels 918 and 1014. L270, “oocytes” not italic.

13. Appendix B, “M. usitatus” should be italic. Appendix C, “maker” should modify to “marker”. Appendix E, “Susceptible stain” should modify to “Susceptible strain”.

14. Reference 12 and 13, The journal name is wrong.

Round 2

Reviewer 3 Report

In their response to my comments, the authors agree that “The current article only speculates that the I873S mutation may affect the Megalurothrips usitatus sodium channel sensitivity to pyrethroids, and the function of this site will be subsequently verified using electrophysiological experiments”.

But instead of doing the actual electrophysiological experiments that would validate or invalidate their hypothesis and perhaps make this study sound, the authors choose to just edit the text and nuance their conclusions.

Functional characterization of I873S and V1015M mutations in the in M. usitatus voltage-gated sodium channel in absence and presence of pyrethroids is crucial. At a minimum, the authors should test the functional effects of these mutations on pyrethroid resistance in another voltage-gated sodium channel already available. Site-directed mutagenesis and electrophysiological experiments on oocytes are straightforward approaches that the authors could either perform or collaborate with other colleagues to perform these important experiments. I see no reason not to perform these experiments and include the results in this study.

Without these experiments, the conclusions of this study remain highly speculative as it is possible that the I873S mutation does not affect the channel sensitivity to pyrethroids at all.

In lines 74-78, the authors claimed: “In this study, the domain II of sodium channel of M. usitatus was cloned, and the mutations that endow resistance to pyrethroid were determined by sequencing. The gene frequency of the obtained mutation was calculated in three field populations of M. usitatus from Hainan, China. Knowing these mutations will help us to quickly understand the resistance of M. usitatus to pyrethroids in the field.

There is no data supporting the hypothesis that any of the mutations identified in this study “endow resistance to pyrethroid”. Therefore, knowing these mutations that only hypothetically endow pyrethroids resistance does not really “help us to quickly understand the resistance of M. usitatus to pyrethroids in the field” unless they are functionally validated.

The data presented in this study do not support such strong statements.

Author Response

We appreciated the reviewer’s comments and suggestions. The purpose of this study is in preparation for subsequent fuller validation of the mutations in relation to M. usitatus pyrethroid resistance, rather than obtaining the exact function of these mutations. Our electrophysiological experiments are already underway and it is expected that the exact function of these mutations will appear in the next article. In addition to the two-electrode voltage clamp technique, it is also intended to use homology modeling and molecular docking to verify the function of these mutations even more. In summary, we hope that we can gain your understanding and we have made changes to the article in response to your comments.

Round 3

Reviewer 3 Report

Despite the fact that the authors agree that it is important to functionally validate I873S mutation as the origin of pyrethroid-resistance in M. usitatus, and the fact that these “electrophysiological experiments are already underway”, the authors decided not to include these important results in this study. As it is currently presented, this study is more speculative than predictive.

Below are my comments:

1)    In the abstract, the authors wrote: “we conducted biological assay and sequenced the voltage gated sodium channel gene from M. usitatus field populations.

This statement is not accurate. Only a small portion of the gene has been sequenced and presented in this study, not the whole gene.

2)    The insecticide bioassay experiments are missing some key controls: test of a different

insecticide for which no resistance has been acquired in these 3 populations.

3)    p-values in table 1 should be defined to indicate what is being compared to what.

Also, additional statistical analysis is important to assess statistical significance of the differences used to interpret the data.

4)    An amino-acid sequence alignment on the regions of the I873S and V1015M mutations should be shown between M. usitatus sodium channels and other sodium channels from other insects and species that have been shown to be sensitive to pyrethroids. This would not only show how conserved amino acids in these positions/regions are but also could perhaps rule out the presence of I873S mutation in pyrethroid sensitive species. Also, this could perhaps make it clear why the authors consider for instance I873S as a mutation in the first place. Is it possible that this position always had Serine even before exposure to pyrethroids?

5)    The amino acid numbering used in this study is not consistent with that of the deposited M. usitatus sodium channel sequence (GeneBank ID UOV21262.1). Foor instance, I873S should be I886S. The authors should fix the numbering in this study to match what is already in the literature.

6)    Line 78-79: “Knowing these mutations will may help us to quickly understand the resistance of M. usitatus to pyrethroids in the field.

“will” should not be here.

Author Response

Reply to the Reviewers about the manuscript earlier submitted as insects-2245642

Reply to Reviewer #3

Despite the fact that the authors agree that it is important to functionally validate I873S mutation as the origin of pyrethroid-resistance in M. usitatus, and the fact that these “electrophysiological experiments are already underway”, the authors decided not to include these important results in this study. As it is currently presented, this study is more speculative than predictive.

RESPONSE: We agree with the reviewer’s comments. We have changed the “predicted to” to “speculated on” in the title.

Below are my comments:

1) In the abstract, the authors wrote: “we conducted biological assay and sequenced the voltage gated sodium channel gene from M. usitatus field populations.” This statement is not accurate. Only a small portion of the gene has been sequenced and presented in this study, not the whole gene

RESPONSE: We agree with the reviewer’s comments. We have added “domain II” after the “gene”.

2) The insecticide bioassay experiments are missing some key controls: test of a different insecticide for which no resistance has been acquired in these 3 populations.

RESPONSE: We appreciated the reviewer’s comments and suggestions. Pyrethroids are the first class of insecticides that act on insect sodium channels, and the aim of this study was also to detect mutation sites in M. usitatus sodium channel that might be involved in pyrethroid resistance mechanisms, so we selected only the commonly used insecticides in this family (lambda-cyhalothrin and deltamethrin), and set indoor susceptible population as controls.

3) p-values in table 1 should be defined to indicate what is being compared to what. Also, additional statistical analysis is important to assess statistical significance of the differences used to interpret the data.
RESPONSE: We appreciated the reviewer’s comments and suggestions. We defined the P-value, see lines 207-208. In addition, the 95% fiducial limit of the LC50 value for the field population is not overlapped with that of the susceptible population, which also proves that the two data sets are significantly different, see lines 124-125.

4) An amino-acid sequence alignment on the regions of the I873S and V1015M mutations should be shown between M. usitatus sodium channels and other sodium channels from other insects and species that have been shown to be sensitive to pyrethroids. This would not only show how conserved amino acids in these positions/regions are but also could perhaps rule out the presence of I873S mutation in pyrethroid sensitive species. Also, this could perhaps make it clear why the authors consider for instance I873S as a mutation in the first place. Is it possible that this position always had Serine even before exposure to pyrethroids?

RESPONSE: We appreciated the reviewer’s comments and suggestions. From a genetic evolutionary point of view, it is possible that serine was present before exposure to the pyrethroids, since isoleucine and serine can be used as alleles, and we speculate that it was pyrethroids stress that caused the susceptible gene (isoleucine) to be screened out, leaving only the resistant gene (serine).

5) The amino acid numbering used in this study is not consistent with that of the deposited M. usitatus sodium channel sequence (GeneBank ID UOV21262.1). Foor instance, I873S should be I886S. The authors should fix the numbering in this study to match what is already in the literature.

RESPONSE: We appreciated the reviewer’s comments and suggestions. The location of the I873S mutation does not correspond to the M. usitatus sodium channel location that has been uploaded on NCBI. Most articles of this type are named after the model insect Drosophila melanogaster sodium channel number, in order to let the reader more directly know whether the mutation has been studied in other insects, and also to be consistent with the numbering of the other two mutations (classical kdr mutations 918 and 1014) detected in this study. We will explain this in the article, please see line 222.

6) Line 78-79: “Knowing these mutations will may help us to quickly understand the resistance of M. usitatus to pyrethroids in the field.” “will” should not be here.

RESPONSE: We agree with the reviewer’s comments. We have deleted “will”.

Round 4

Reviewer 3 Report

The authors did not fully address comment #4 from my previous review comments, i.e. showing an amino-acid sequence alignment on the regions of the I873S and V1015M mutations between M. usitatus sodium channels and other sodium channels from other insects and species that have been shown to be sensitive to pyrethroids. This would not only show how conserved amino acids in these positions/regions are but also could perhaps rule out the presence of I873S mutation in pyrethroid sensitive species. If that’s the case, this would perhaps provide some rationale on why the authors consider for instance I873S as a mutation in the first place, especially in the absence of functional validation.

Author Response

Reply to the Reviewers about the manuscript earlier submitted as insects-2245642

Reply to Reviewer #3

The authors did not fully address comment #4 from my previous review comments, i.e. showing an amino-acid sequence alignment on the regions of the I873S and V1015M mutations between M. usitatus sodium channels and other sodium channels from other insects and species that have been shown to be sensitive to pyrethroids. This would not only show how conserved amino acids in these positions/regions are but also could perhaps rule out the presence of I873S mutation in pyrethroid sensitive species. If that’s the case, this would perhaps provide some rationale on why the authors consider for instance I873S as a mutation in the first place, especially in the absence of functional validation.

RESPONSE: We agree with the reviewer’s comments. I am very sorry for misunderstanding your point, as you said, the isoleucine on 873 of the sodium channels in sensitive strains of Megalurothrips usitatus, Frankliniella occidentalis and Thrips palmi, but serine in the M. usitatus resistant strains (Appendix D), could indeed demonstrate that I873S is a novel mutation site that may be involved in the pyrethroid resistance mechanism. We have added some descriptions in the abstract, please see lines 28-29.